# Understanding the Enhanced Protective Mechanism of CoCrNiAlY–YSZ–LaMgAl$_{11}$O$_{19}$ Double-Ceramic Coating with Aluminum Plating

**Junfei Xu** [†], **Zhiguo Wang** [†], **Shuai Hu** [†], **Yongjun Feng** *, **Suying Hu, Yongjun Chen** * **and Zhiwen Xie**

School of Mechanical Engineering and Automation, University of Science and Technology Liaoning, Anshan 114051, China; xjfword2021@163.com (J.X.); w13604864782@126.com (Z.W.); 17805480960@163.com (S.H.); ying5215116@163.com (S.H.); xzwustl@126.com (Z.X.)
* Correspondence: xfsword@163.com (Y.F.); chenyongjun-net@163.com (Y.C.); Tel.: +86-412-592-9790 (Y.F. & Y.C.)
† The authors contributed equally to this work.

**Abstract:** To understand the enhanced protection mechanism of CoCrNiAlY–YSZ–LaMgAl$_{11}$O$_{19}$ double-layer ceramic coating with aluminum plating, a finite element simulation method was used to simulate the distribution of thermal stress in the coating in all directions. The results show that in the air exposure of the un-aluminized coating, high temperature causes a large radial thermal stress on the surface of the LaMgAl$_{11}$O$_{19}$ (LMA) layer, and it increases with the increase in temperature, which is the main reason for the initiation of axial cracks. After arc aluminum plating, the aluminum plating layer effectively inhibited the volume shrinkage of the coating through good adhesion to the coating and internal diffusion; the thermal stress of the coating was considerably reduced; and the CoCrNiAlY–YSZ–LMA coating had an effective enhancement and protection effect. However, there was still a certain amount of shear thermal stress inside the LMA layer, the top of the crack, and the bottom of the crack. This thermal stress caused the initiation of radial microcracks in the LMA layer, which also becomes a risk point for the failure of the aluminum coating.

**Keywords:** double-ceramic coating; arc aluminum plating; stress; protection mechanism; finite element simulation



## 1. Introduction

The working temperature of aeroengines has increased, with the inlet temperature of the turbine reaching values as high as 1750 °C [1]. The hot-end parts made of conventional nickel-based superalloys work at high temperatures for a long time, which leads to a reduction in their oxidation resistance and heat corrosion resistance [2]. To solve this problem, thermal barrier coatings (TBCs) have been developed for the surface protection of superalloys [3–7]. However, when the working temperature of the TBC exceeds 1200 °C, the coating undergoes significant phase change and volume change, triggering high internal stress and initiating cracks, which eventually lead to premature fracture failure [8]. Therefore, it is necessary to develop new TBC materials with better heat resistance and higher thermal stability.

Chen et al. found that LaMgAl$_{11}$O$_{19}$ (LMA) coating can still work well at 1250 °C, being a promising TBC material [9], and that the LaMgAl$_{11}$O$_{19}$/YSZ dual ceramic coating can alleviate the thermal stress concentration of the substrate, improve the bonding strength of the coating and the substrate, and perfectly overcome the shortcomings of traditional single ceramic coatings [10]. These materials enable feasible methods and strategies for the structural design of TBCs [11–13]. We found in previous research that the CoCrNiAlY–YSZ–LaMgAl$_{11}$O$_{19}$ double ceramic coating provides a significant improvement in the thermal corrosion resistance of the coating at a high temperature of 1200 °C, but the volume shrinkage of the coating leads to the initiation of a large number of microcracks. These microcracks act as oxidation diffusion channels, leading to the deterioration of the

barrier effect of the coating [14]. To improve the oxidation resistance of the CoCrNiAlY–YSZ–LaMgAl$_{11}$O$_{19}$ coating, we developed a new type of arc aluminum plating process. Experimental studies have shown that the aluminum-plated layer is effectively bonded to the LMA layer, inhibiting its volume shrinkage and the initiation of large-size axial microcracks. Moreover, at high temperatures, the Al$_2$O$_3$ will melt and fill into the LMA cracks; the synergistic effect of the dense ceramic layer and Al$_2$O$_3$ layer effectively inhibit the inner diffusion of oxygen, consequently reducing the rapid TGO layer growth, achieving good oxidation resistance and suppressing the severe fracturing of the interface between the CoCrNiAlY and YSZ layers [15].

Through finite element analysis, the distribution of the thermal stress of the coating can be revealed in detail, and the enhanced protection mechanism of the arc aluminum coating on the CoCrNiAlY–YSZ–LaMgAl$_{11}$O$_{19}$ double ceramic coating can be understood more accurately [16–23]. Therefore, the finite element simulation method was adopted in this study to analyze in detail the thermal stress distribution of CoCrNiAlY–YSZ–LaMgAl$_{11}$O$_{19}$ coating at 900 °C, 1000 °C, 1100 °C, and 1200 °C, before and after arc aluminum plating. This, this study aimed to understand and analyze the relationship between thermal stress and crack initiation and propagation and provide detailed data as well as more effective strategies for the design and service behavior study of CoCrNiAlY–YSZ–LaMgAl$_{11}$O$_{19}$ dual ceramic coatings.

## 2. Material and Methods

### 2.1. Coating Preparation

Using the GH199 superalloy (Fushun Special Steel Shares Co., Ltd., Fushun, China) as the matrix, two specimens were prepared: one was a CoCrNiAlY–YSZ–LaMgAl$_{11}$O$_{19}$ double ceramic coating specimen (denoted as M1) prepared by atmospheric pressure plasma spraying [24] (Multicat, Oerlikon Metco, Swizerland), and the other was a CoCrNiAlY–YSZ–LaMgAl$_{11}$O$_{19}$ double ceramic coating specimen (denoted as M2) with an aluminized surface (99.9% pure aluminum). The length, width and height of the two specimens were 15 mm, 15 mm and 5 mm. Before spraying, the samples underwent three steps: sandpaper grinding to remove oxides, alcohol ultrasonic cleaning and sandblasting. Table 1 shows the spraying powder composition and the spraying parameters of each layer in the two test pieces. With high-purity Al (99.9%) as the arc target, the deposition parameters of the AIP method when depositing Al were as follows: current 80 A, argon pressure 1.0 Pa, −80 bias, and deposition time 60 min. The surface and cross-sectional structures of all coatings were characterized by scanning electron microscopy (SEM, Zeiss ΣIGMA HD, Carl Zeiss, Jena, Germany) to provide coating structure parameters for the establishment of finite element analysis models.

**Table 1.** Spraying powder composition and spraying parameters of each layer.

| Material | Element | Current/A | Distance/mm | Line Speed/mm/min | Particle Diameter/μm |
|---|---|---|---|---|---|
| CoCrNiAlY | 31%–34% Ni, 24.5%–26.5% Cr, 5.0%–6.5% Al, 0.4%–0.8% Y and balanced Co | 500 | 120 | 800 | 30–74 |
| YSZ | 7.0%–7.5% Y$_2$O$_3$ and balanced ZrO$_2$ | 600 | 120 | 800 | 30–64 |
| LMA | 15.0%–24.0% La$_2$O$_3$ and 4.0%–7.0% MgO, and balanced Al$_2$O$_3$ | 600 | 120 | 800 | 32–125 |

### 2.2. Finite Element Simulation

After the CoCrNiAlY–YSZ–LaMgAl$_{11}$O$_{19}$ double ceramic coating was arc-plated with aluminum, there were two types of combinations of the aluminum-plated layer and the coating: in the first, the aluminized layer covered the surface of the LMA layer, which was defined as "arc aluminized surface coverage" (denoted as M2S1); in the other, the aluminized layer was filled into the cracks in the axial direction of the coating (the direction perpendicular to the crack surface), which was defined as "arc aluminized crack filling" (denoted as M2S2). For a comparative study, three finite element models were established, and the corresponding boundary conditions and loads were defined. Table 2 lists the corresponding layer structures of the three finite element models.

**Table 2.** The layer structure corresponding to the finite element model.

| Finite Element Model | Coating Structure | Combination Form of Aluminized Layer |
| :---: | :---: | :---: |
| M1 | CoCrNiAlY + YSZ + LMA | — |
| M2S1 | CoCrNiAlY + YSZ + LMA + Al plating layer | Surface coverage |
| M2S2 | CoCrNiAlY + YSZ + LMA + Al plating layer | Crack filling |

### 2.2.1. Finite Element Model

According to the SEM images of the cross-sectional structures of specimens M1 and M2 (see literature [9]), the three-dimensional physical model shown in Figure 1 was established. To facilitate the analysis and calculation, this model was simplified to a two-dimensional physical model, and the right half was used for analysis to establish three two-dimensional physical models, as shown in Figure 2. The distance from the center of symmetry of the model to the edge of the model in Figure 2 is represented by the X coordinate, which is defined as the radial direction, and the distance from the bottom of the substrate to the surface layer is represented by the Y coordinate, which is defined as the axial direction. An Abaqus four-node axisymmetric temperature displacement coupled quadrilateral element (CAX4T) was used for the meshing and simulation calculations. The thickness (axial) and width (radial) dimensions of each layer in the finite element model are shown in Table 3; the axial crack of the coating was simplified to a rectangle with a width of 5 μm and a depth of 40 μm. The distance from the axis of symmetry was based on the analysis results to determine the position of the maximum thermal stress.

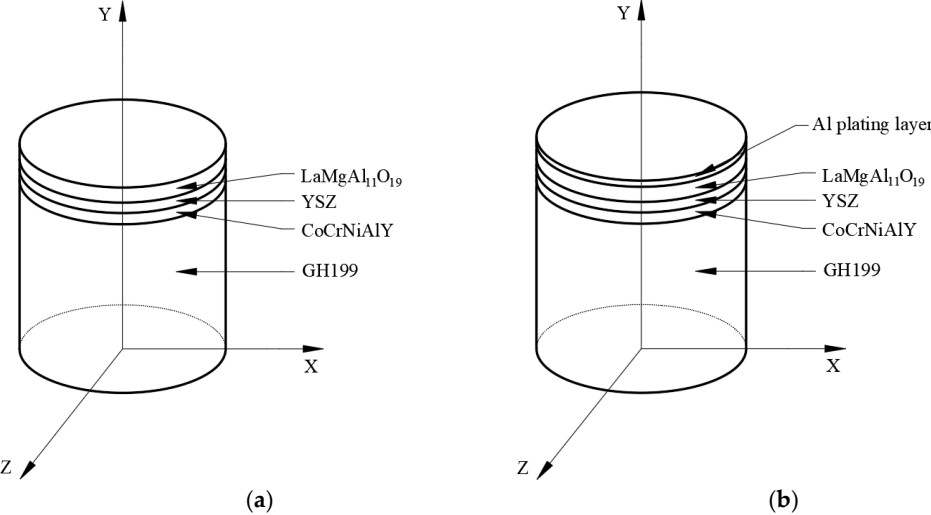

**Figure 1.** Three-dimensional finite element physical model: (**a**) M1, (**b**) M2.

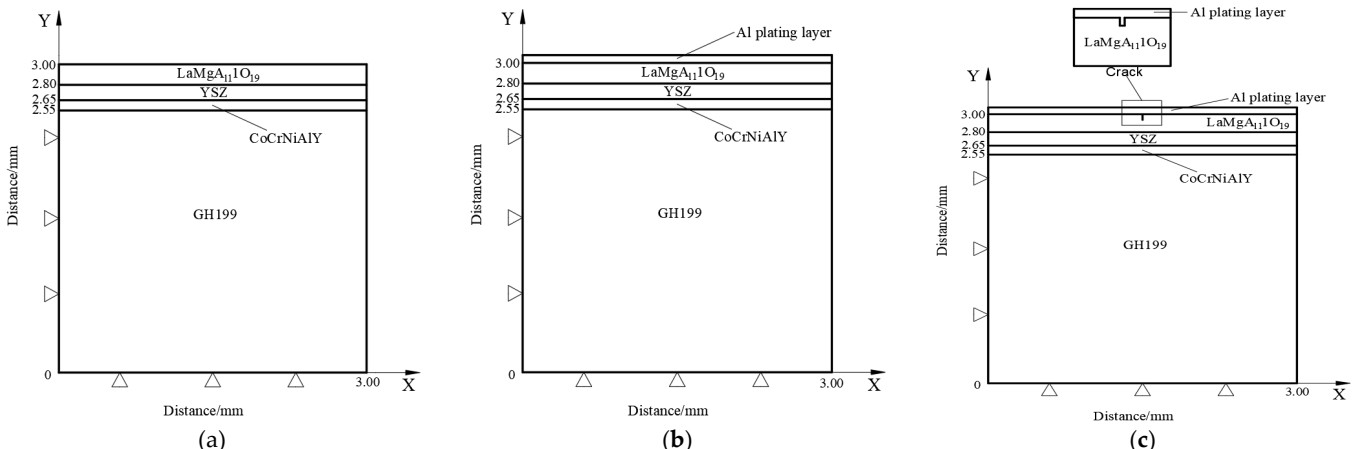

**Figure 2.** Two-dimensional finite element physical model: (**a**) M1, (**b**) M2S1, (**c**) M2S2.

**Table 3.** Thickness and width of each layer in the finite element model.

| Finite Element Model | Coating Thickness/μm | | | | | Coating Width/μm |
|---|---|---|---|---|---|---|
| | GH199 | CoCrNiAlY | YSZ | LaMgAl$_{11}$O$_{19}$ | Al Plating Layer | |
| M1 | 2550 | 100 | 150 | 200 | 20 | 3000 |
| M2S1 | 2550 | 100 | 150 | 200 | 20 | 3000 |
| M2S2 | 2550 | 100 | 150 | 200 | 20 | 3000 |

2.2.2. Material Parameters and Basic Assumptions

The elastic modulus E, thermal expansion coefficient $\alpha$, thermal conductivity $\lambda$, specific heat capacity C, density $\rho$, and Poisson's ratio $\upsilon$ of the material are listed in Table 4 [25–33]. To facilitate the calculation, the following assumptions were considered in the model:

1.  The residual stress of the coating (including the arc aluminum coating) and the substrate at the initial temperature were zero;
2.  The entire model was isotropic;
3.  The model had no plastic failure, the bonding between the coatings was firm, and there was no relative sliding.

**Table 4.** Material parameters of coating [25–33].

| Coatings | T/°C | E/GPa | $\alpha$/(10$^{-6}$ K$^{-1}$) | $\lambda$/(Wm$^{-1}$·°C$^{-1}$) | C/(J·kg$^{-1}$·°C$^{-1}$) | $\rho$/(kg·m$^{-3}$) | $\upsilon$ |
|---|---|---|---|---|---|---|---|
| GH199 | 25 | 205 | 12.1 | 13.38 | 372.6 | 8260 | 0.30 |
| | 100 | 203 | 12.2 | 13.68 | 372.8 | 8260 | 0.30 |
| | 300 | 193 | 13.4 | 20.27 | 456.4 | 8260 | 0.30 |
| | 500 | 180 | 14.3 | 24.62 | 452.2 | 8260 | 0.30 |
| | 700 | 166 | 15.5 | 29.05 | 515.0 | 8260 | 0.30 |
| | 900 | 149 | 16.1 | 33.44 | 561.0 | 8260 | 0.30 |
| | 1000 | 136 | 15.4 | 33.37 | 581.9 | 8260 | 0.30 |
| | 1100 | 124 | 14.8 | 33.30 | 607.0 | 8260 | 0.30 |
| | 1200 | 112 | 14.1 | 33.21 | 627.9 | 8260 | 0.30 |
| CoCrNiAlY | 25 | 225 | 14 | 4.3 | 501 | 7320 | 0.30 |
| | 400 | 186 | 24 | 6.4 | 592 | 7320 | 0.30 |
| | 800 | 147 | 47 | 10.2 | 781 | 7320 | 0.30 |
| | 1200 | 90 | 71 | 16.1 | 764 | 7320 | 0.30 |
| YSZ | 20 | 48 | 10.4 | 1.80 | 450 | 5280 | 0.10 |
| | 200 | 47 | 10.5 | 1.76 | 491 | 5280 | 0.10 |
| | 500 | 43 | 10.7 | 1.75 | 532 | 5280 | 0.10 |
| | 700 | 39 | 10.8 | 1.72 | 573 | 5280 | 0.10 |
| | 1100 | 25 | 10.9 | 1.69 | 615 | 5280 | 0.10 |
| | 1200 | 22 | 11.0 | 1.67 | 656 | 5280 | 0.10 |

**Table 4.** *Cont.*

| Coatings | T/°C | E/GPa | $\alpha/(10^{-6}\,K^{-1})$ | $\lambda/(Wm^{-1}\cdot°C^{-1})$ | $C/(J\cdot kg^{-1}\cdot°C^{-1})$ | $\rho/(kg\cdot m^{-3})$ | $\upsilon$ |
|---|---|---|---|---|---|---|---|
| LMA | 20 | 28.83 | 8.3 | 1.53 | 578.4 | 3321 | 0.23 |
| | 200 | 25.47 | 9.5 | 1.18 | 805.4 | 3321 | 0.23 |
| | 400 | 22.11 | 10.5 | 0.82 | 913.2 | 3321 | 0.23 |
| | 600 | 18.75 | 11.0 | 0.65 | 1007.9 | 3321 | 0.23 |
| | 800 | 15.37 | 11.5 | 0.52 | 1055.3 | 3321 | 0.23 |
| | 1000 | 12.01 | 12.0 | 0.41 | 1089.6 | 3321 | 0.23 |
| | 1200 | 8.65 | 13.0 | 0.32 | 1094.5 | 3321 | 0.23 |
| Al plating layer | 20 | 400 | 8 | 10 | 1000 | 3500 | 0.23 |
| | 200 | 390 | 8.2 | 7.794 | 1000 | 3500 | 0.23 |
| | 400 | 380 | 8.4 | 6.029 | 1000 | 3500 | 0.24 |
| | 600 | 370 | 8.7 | 5.074 | 1000 | 3500 | 0.24 |
| | 800 | 355 | 9 | 4.412 | 1000 | 3500 | 0.25 |
| | 1000 | 325 | 9.3 | 4.412 | 1000 | 3500 | 0.25 |
| | 1100 | 320 | 9.6 | 4 | 1000 | 3500 | 0.25 |

### 2.2.3. Boundary and Initial Conditions

1.  Force boundary conditions: the degree of freedom in the X direction at the symmetry axis of the model U1 = 0, and the degree of freedom in the Y direction at the bottom of the model U2 = 0.
2.  Thermal boundary conditions: the upper end of the model was exposed to air, and convective heat transfer occurred. The convection coefficient was 65 W/(°C·m$^2$), and the left, right, and lower end faces were heated.
3.  Initial conditions: the initial temperature of the model was equal to the ambient temperature (25 °C), and the model was in an unstressed state.

### 2.2.4. Temperature Load

Four maximum operating temperatures of 900 °C, 1000 °C, 1100 °C, and 1200 °C were set, and 1200 °C was consistent with the experimental study. Four temperature displacement coupling analysis steps I, II, III, and IV were established for each maximum operating temperature. The coupled analysis period was 6 h. At t = 0 h, the model was at ambient temperature (25 °C), and the temperature was increased to the highest working temperature within 1.5 h and maintained for 2 h; then, the temperature was reduced to room temperature (25 °C) within 1.5 h, and maintained for 1 h (see Figure 3 for details).

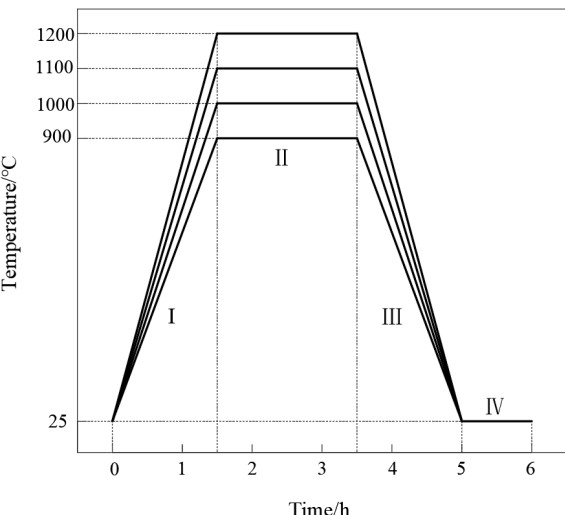

**Figure 3.** Four coupled analysis steps of temperature displacement at the four highest operating temperatures.

### 2.3. Experimental Study

A high-temperature muffle furnace was used to study the oxidation behavior of the M1 and M2 coating specimens in air at a working temperature of 1200 °C, with the

oxidation time of 10 h, and heating rate of 10 °C/min. A scanning electron microscope (SEM, Zeiss ΣIGMA HD) was used to characterize the cross-sectional structures of the coatings, and an X-ray diffractometer was used to analyze the phase structure before and after oxidation. Through the XRD results, the specific properties of the coating material could be accurately understood to facilitate the establishment of the finite element model. The initiation and propagation of coating cracks were analyzed through SEM images for comparison with the finite element simulation results and the enhanced protection mechanism of the arc aluminum plating on CoCrNiAlY-YSZ-LMA double ceramic coating. The working temperature, time and heating rate settings of the oxidation test are detailed in [9].

### 3. Results

*3.1. M1, M2S1 Simulation*

According to the regulations of the Abaqus system, the radial stresses S11 and S22 are positive for tensile stress and negative for compressive stress. The positive value of the shear stress S12 is the stress in the XY plane along the positive direction of Y, and the negative value is the opposite. According to Figure 4a, the axial thermal stress and shear thermal stress on the surface of the LMA layer at a working temperature of 900 °C before aluminizing are close to zero, but there is a large radial thermal stress (tensile stress, the direction is perpendicular to Y), and its maximum value is 0.233 GPa. The radial thermal stress is considered the main cause of the coating axial (Y-direction) crack initiation, which is a type I crack. According to Figure 4b, it can be seen that the axial thermal stress on the surface of the LMA layer at the working temperature of 900 °C after aluminum plating approaches zero. The maximum radial thermal stress is 0.107 GPa, which is significantly lower than that before aluminizing. It can be concluded that the aluminized layer and the LMA layer are well combined, which effectively reduces the radial thermal stress and inhibits the initiation of axial cracks in the coating. At the same time, it was found that a certain shear thermal stress appeared after aluminum plating, and it was along the negative direction of Y. This shows that the aluminum-plated layer inhibited the surface shrinkage of the coating by adhering to the surface of the LMA layer, but the inhibition ability gradually decreased along the depth direction. The shrinkage of the coating releases downward shear thermal stress, which can cause radial microcracks (X direction).

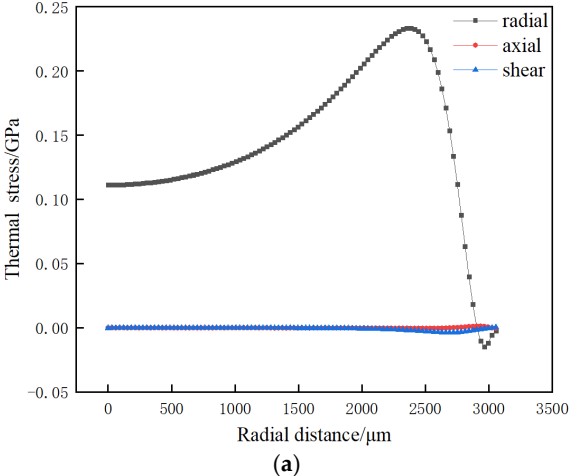
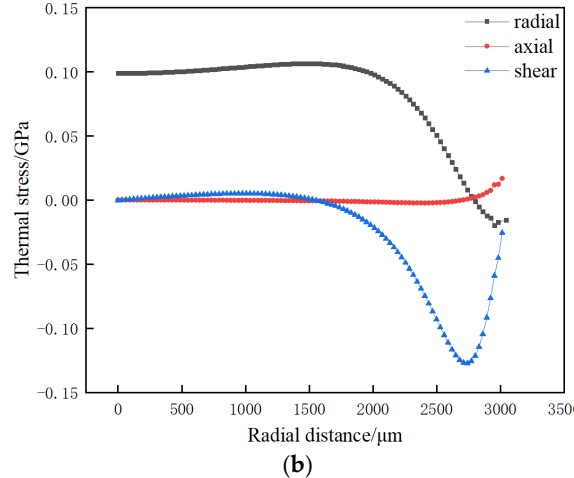

**Figure 4.** Distribution of thermal stress along the X direction on the surface of the LMA layer of the (**a**) M1 and (**b**) M2S1 specimens at a working temperature of 900 °C.

Moreover, the simulation analysis at the 1000 °C, 1100 °C, and 1200 °C working temperatures was continued, focusing on the radial thermal stress and based on the above analysis results. The data in Table 5 and the curve in Figure 5a shows that the maximum radial thermal stress on the surface of the LMA layer increased from 0.233 GPa at 900 °C

to 0.254 GPa at 1200 °C under the four operating temperatures of the M1 un-aluminized specimen. Thus, the volume shrinkage of the LMA layer increased with the increase in operating temperature, resulting in gradually increasing thermal stress, leading to increasingly serious axial (Y direction) microcracks. The data in Table 5 and the curve in Figure 5b accurately indicate that the maximum radial thermal stress on the surface of the LMA layer of the M2 S1 aluminized specimen was reduced from 0.107 GPa at 900 °C to 0.091 GPa at 1200 °C at the four working temperatures. This indicates that the thermal stress of the LMA layer surface was reduced under the effective adhesion of the aluminum layer, and the volume shrinkage of the LMA layer was suppressed. Moreover, the higher the working temperature, the stronger the suppression effect, which can effectively prevent the initiation of axial (Y direction) microcracks on the surface of the LMA layer.

**Table 5.** Maximum radial thermal stress value of the LMA layer surface of the M1 and M2S1 specimens at the four operating temperatures.

| Maximum Radial Tensile Stress/Gpa | Temperature Load/°C | | | |
| :---: | :---: | :---: | :---: | :---: |
| | **900** | **1000** | **1100** | **1200** |
| M1 | 0.233 | 0.252 | 0.258 | 0.254 |
| M2S1 | 0.107 | 0.11 | 0.105 | 0.091 |

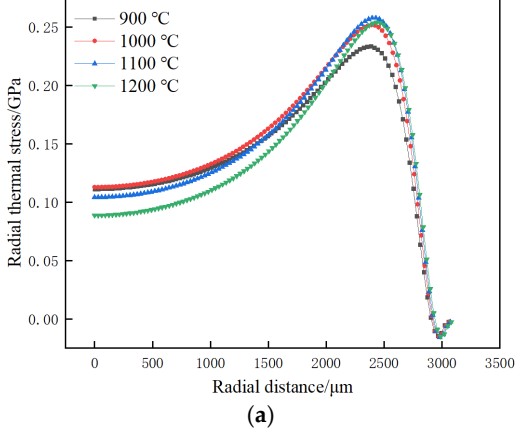
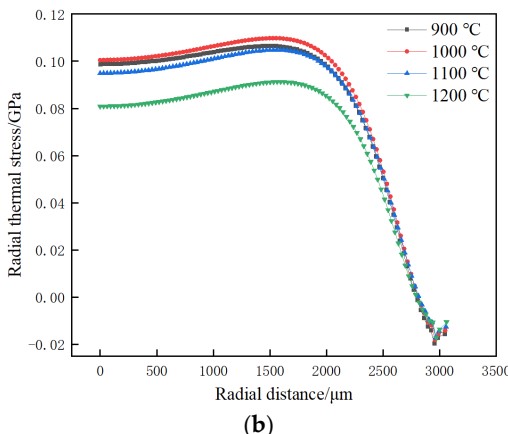

**(a)**　　　　　　　　　　　　　　　　**(b)**

**Figure 5.** Comparison of radial thermal stress distribution on the surface of LMA layer of (**a**) M1 and (**b**) M2S1 specimens at four operating temperatures.

From Figure 6a, the maximum radial thermal stress on the surface of the LMA layer of the M1 specimen was positively correlated with the working temperature, increasing with an increase in the working temperature. Figure 6b shows that the maximum radial thermal stress on the surface of the LMA layer of the M2S1 specimen was negatively correlated with the operating temperature, decreasing as the operating temperature increased. At the four working temperatures, the aluminum-plated layer effectively reduced the radial thermal stress on the surface of the LMA layer, and the higher the working temperature, the larger the reduction. Among the temperatures analyzed, the working temperature of 1200 °C was 64% lower than that before aluminizing, which indicates that the aluminized layer has a good protective effect.

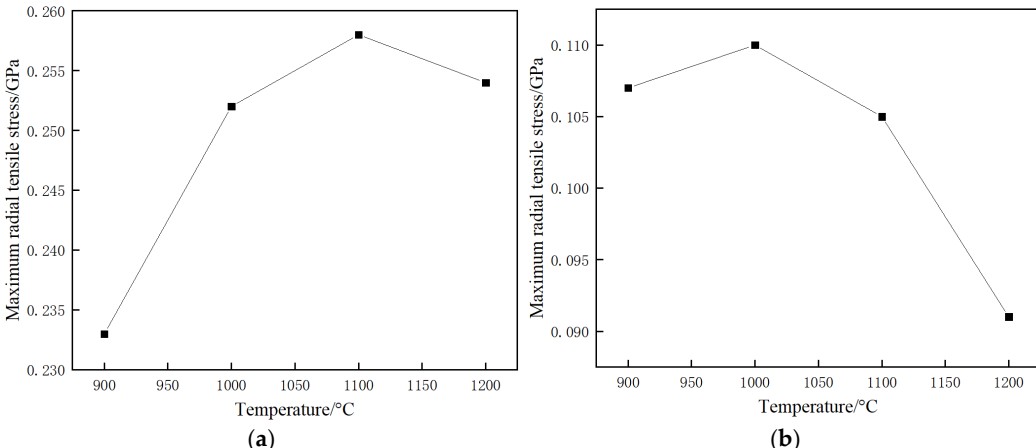

**Figure 6.** Relationship between the maximum radial thermal stress on the surface of the LMA layer of the (**a**) M1 and (**b**) M2S1 specimens and the operating temperatures.

Similar to the working temperature of 900 °C, a certain shear thermal stress appeared on the surface of the LMA layer of the aluminum-plated specimen at the other three working temperatures. Figure 7 shows that the maximum downward shear thermal stresses generated at the working temperatures of 900 °C, 1000 °C, 1100 °C, and 1200 °C were 0.127 GPa, 0.143 GPa, 0.155 GPa, and 0.168 GPa, respectively, which were positively correlated with the operating temperature (see Figure 8). This indicates that the aluminum-plated layer inhibited the shrinkage of the coating surface by adhering to the LMA layer, and the shrinkage inside the coating was also inhibited. However, along the depth direction, the restraining ability gradually decays, the volume shrinkage intensifies, and the downward shear thermal stress is released; thus, radial microcracks are generated inside the LMA. The higher the operating temperature, the more severe the radial microcrack initiation.

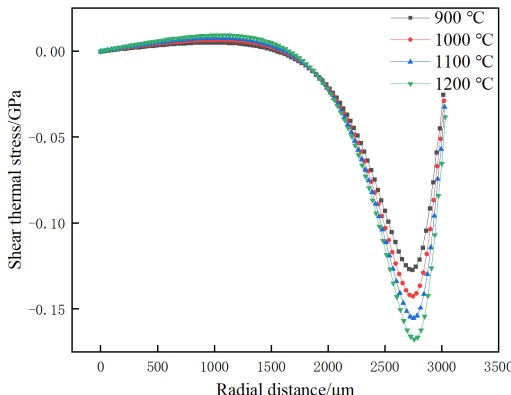

**Figure 7.** Comparison of the shear thermal stress distribution on the surface of the LMA layer of the M2S1 specimen at four operating temperatures.

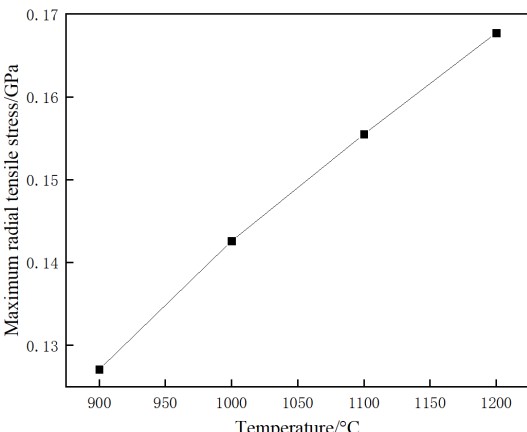

**Figure 8.** Relationship between the maximum negative shear thermal stress on the surface of the LMA layer of the M2S1 specimen and the working temperature.

*3.2. M2S2 Simulation*

According to Figure 4b, the maximum radial thermal stress on the surface of the LMA layer after aluminum plating appears at 1500 μm on the right side of the Y-axis; thus, the axial crack model was established accordingly. As shown in Figure 2c, the upper opening of the crack was at 3000 μm in the Y direction, and the bottom of the crack was at 2960 μm in the Y direction. The working temperature was set to 1200 °C, and the stress distribution of the sidewalls of the axial cracks filled with aluminum plating on the M2S2 specimens under the two conditions of "maintaining at 1200 °C" and "reducing to room temperature" was analyzed.

Figure 9 shows the left and right sidewalls at the crack (except for the four points a, b, c, and d at the crack opening and the bottom of the crack (see Figure 10)). The thermal stress values in all directions approached zero, which indicates that the aluminum plating was well bonded to the crack sidewall after filling the crack, effectively inhibiting the shrinkage of the coating at the crack, preventing further expansion of the crack, and also blocking the passage of oxygen into the crack. However, according to the data in Table 6, the maximum value of the isotropic stress appears near the four points a, b, c, d on the left and right of the crack opening and the bottom of the crack. This shows that at a working temperature of 1200 °C, the aluminum-plated layer had a stronger inhibitory effect on the crack propagation at the top and bottom; thus, this is the dangerous point for failure of the aluminum-plated layer, which can be confirmed from the shear thermal stress cloud diagram shown in Figure 10.

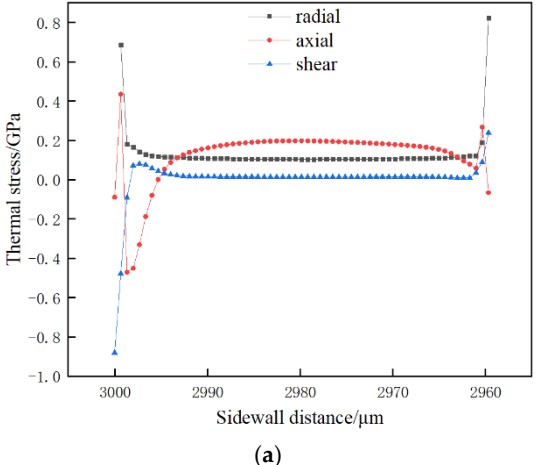

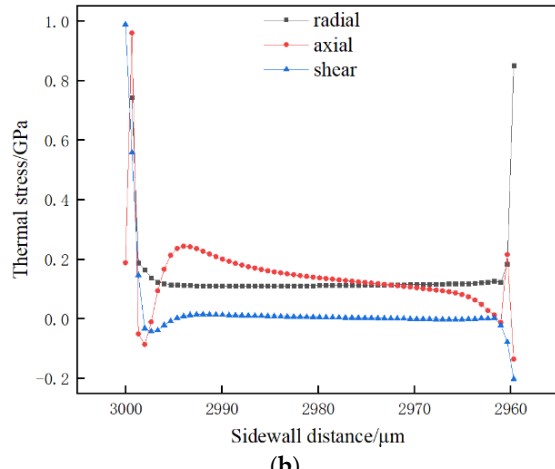

(**a**)  (**b**)

**Figure 9.** Distribution of thermal stress along the depth of the sidewall, (**a**) Left sidewall, (**b**) Right sidewall of the M2S2 crack at the "maintained at 1200 °C" situation.

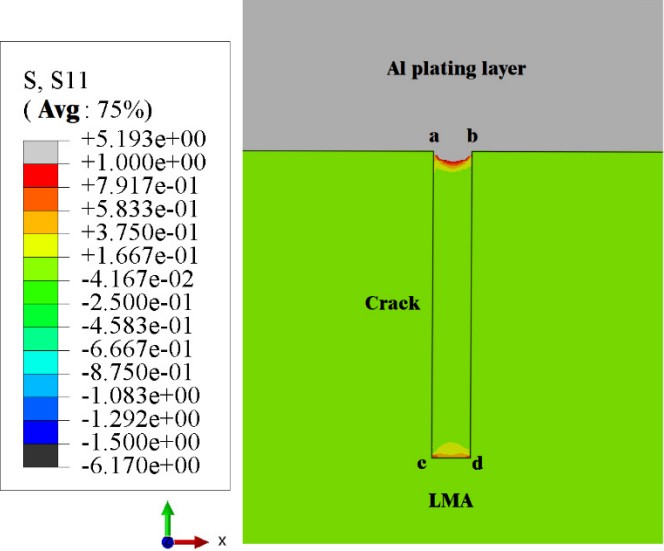

**Figure 10.** Shear thermal stress cloud diagram at the top and bottom of M2S2 cracks "maintained at 1200 °C".

**Table 6.** Maximum and minimum thermal stresses in each direction on the sidewall of M2S2 cracks at the "maintained at 1200 °C" situation.

| Maximum Thermal Stress/Gpa | Radial | Axial | Shear |
|---|---|---|---|
| Left side wall | 0.82 | 0.44 | 0.24 |
|  | 0.1 | −0.47 | −0.09 |
| Right side wall | 0.85 | 0.95 | 0.99 |
|  | 0.11 | −0.14 | −0.2 |

The situation of "reducing to room temperature" was further analyzed at a working temperature of 1200 °C. It can be seen from Figure 11 that after "reducing to room temperature," there was still a certain degree of residual thermal stress on the left and right sides of the crack, and its distribution was the same as that under the condition of "maintaining at 1200 °C." Moreover, the effect on the crack was also the same. However, it can be seen from Table 7 that the extreme values of the residual thermal stress in all directions slightly increased, which further shows that the upper and bottom of the crack are dangerous points for the failure of the aluminum coating.

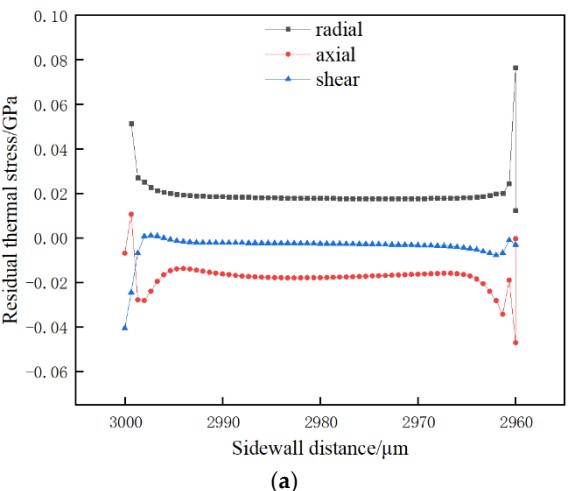

(**a**)

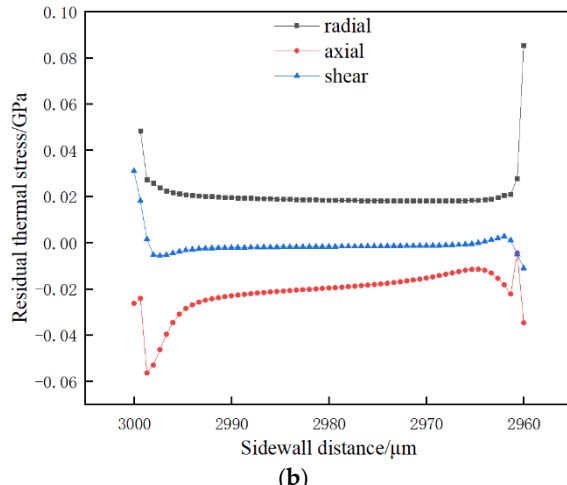

(**b**)

**Figure 11.** Distribution of residual thermal stress along the depth of the sidewall (**a**) Left sidewall, (**b**) Right sidewall of the M2S2 crack at the "reduce to room temperature" situation.

**Table 7.** Maximum and minimum residual thermal stresses on the sidewalls of M2S2 cracks at the "reduced to room temperature" situation.

| Maximum Residual Thermal Stress/Gpa | Radial | Axial | Shear |
|---|---|---|---|
| Left side wall | 0.07 | 0.01 | 0.001 |
| | 0.01 | −0.04 | −0.04 |
| Right side wall | 0.085 | −0.004 | 0.03 |
| | 0.02 | −0.06 | −0.01 |

### 3.3. Experimental Analysis

Figure 12 shows the SEM cross-sectional images of all the coatings after the oxidation test at a working temperature of 1200 °C. The aluminized layer had a significant inhibitory effect on the crack initiation and propagation of the CoCrNiAlY-YSZ-LMA dual-ceramic coating. It can be seen from Figure 12a,b a large number of axial (Y-direction) microcracks in the double ceramic coating LMA layer without aluminum plating; some of the microcracks penetrated the coating to become oxygen transport channels, accelerating the rapid growth of the TGO layer at the YSZ–CoCrNiAlY interface and resulting in severe interface fracture. The above experimental results are highly consistent with the results of the M1 simulation. Therefore, it can be inferred that the high temperature causes a serious shrinkage of the coating volume, which causes a large radial thermal stress on the LMA surface, which in turn, induces a large number of axial microcracks.

After arc aluminum plating, the axial microcracks on the surface of the LMA were essentially eliminated, as seen from Figure 12c,d, but radial microcracks (Y direction) were generated in the LMA. This result is consistent with the M2S1 simulation analysis results. Therefore, it can be inferred that the aluminum coating has a better protective role. The protection mechanism is as follows: the aluminized layer adheres to the LMA surface, inhibits its volume shrinkage, reduces the radial thermal stress on the surface of the LMA layer, and effectively prevents the initiation of axial microcracks. However, owing to the downward Y direction of the LMA layer, the inhibitory effect of the aluminum coating gradually attenuates, and the volume shrinkage gradually intensifies, resulting in a certain shear thermal stress (Y direction), and causing radial cracks in the LMA layer.

After arc aluminum plating, the aluminum plating layer completely melted and filled into the existing axial microcracks under the action of high temperature, as observed in Figure 12e,f; thus, the cracks were healed, and the oxygen diffusion channel was blocked. This result is consistent with the M2S2 simulation analysis results. Therefore, it can be inferred that the aluminum plating layer has a better enhancement effect. The enhancement mechanism is as follows: the aluminum-plated layer adheres to the crack sidewalls, eliminates the all-directional thermal stress on the crack sidewall, and prevents further crack propagation. The oxygen diffusion channel is blocked, reducing the rapid growth of the TGO layer, thereby inhibiting interface fracture (see Figure 12e).

Figure 13 shows the XRD results of the Al coating surface before and after the oxidation test at 1200 °C. Before the oxidation test, the XRD results were in good agreement with the coating preparation plan. The sample M1 was completely dominated by the LMA phase, and M2 was completely dominated by the Al phase. After the oxidation test, there was a trace of $LaAlO_3$ phase + $Al_2O_3$ phase in sample M1. According to previous research results, it can be known that this is obtained by high temperature decomposition of LMA phase. The $LaAlO_3$ phase and a small amount of $Al_2O_3$ phase in the sample M2 are the high-temperature decomposition products of the LMA phase, and most of the $Al_2O_3$ phase is obtained by high-temperature oxidation of the Al layer. In the model establishment, the error caused by a small amount of $LaAlO_3$ phase is considered to be very small, so the model establishment is more consistent with the experimental XRD results.

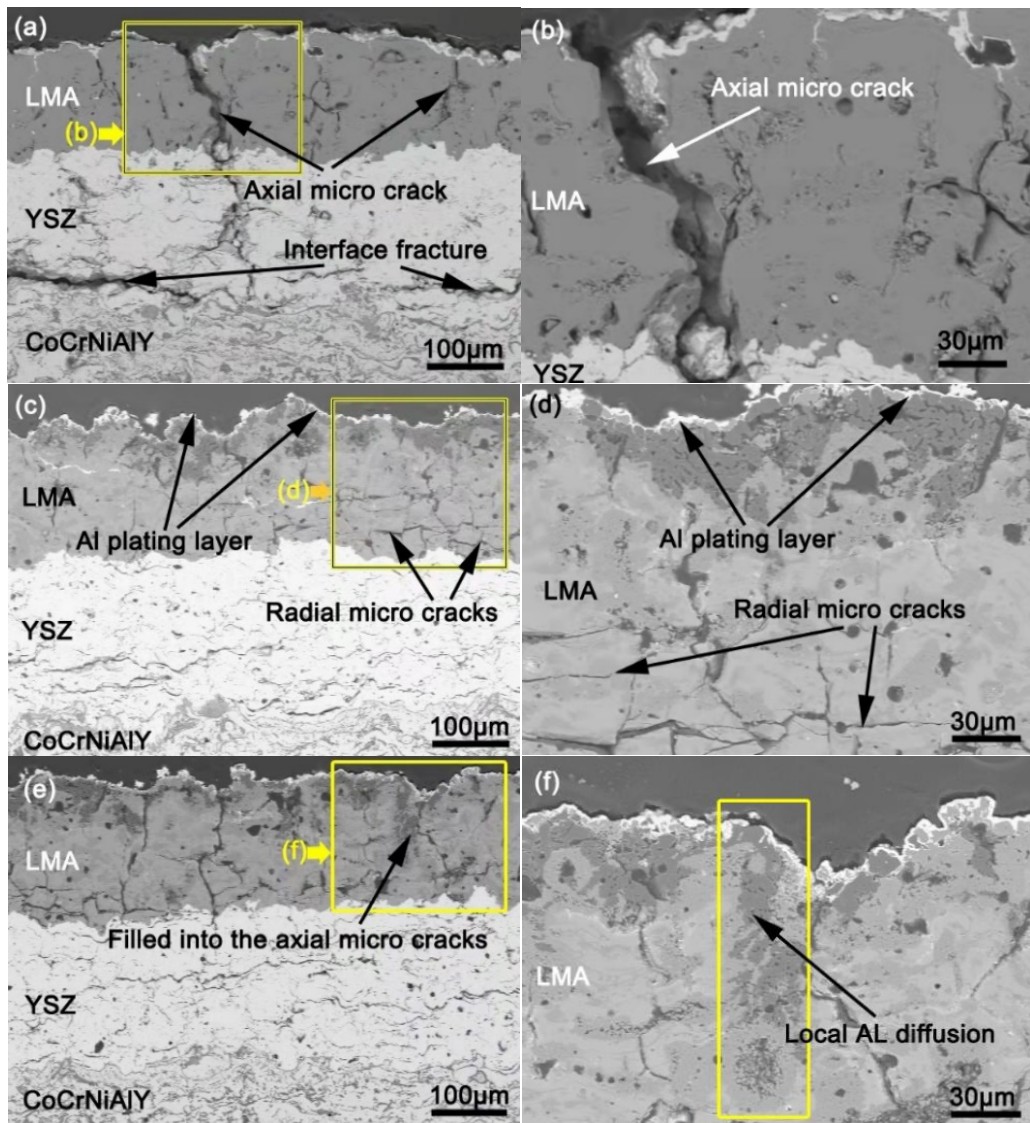

**Figure 12.** SEM cross-sectional images of all coatings after oxidation test at 1200 °C working temperature: (**a**,**b**) M1; (**c**,**d**) M2S1; (**e**,**f**) M2S2.

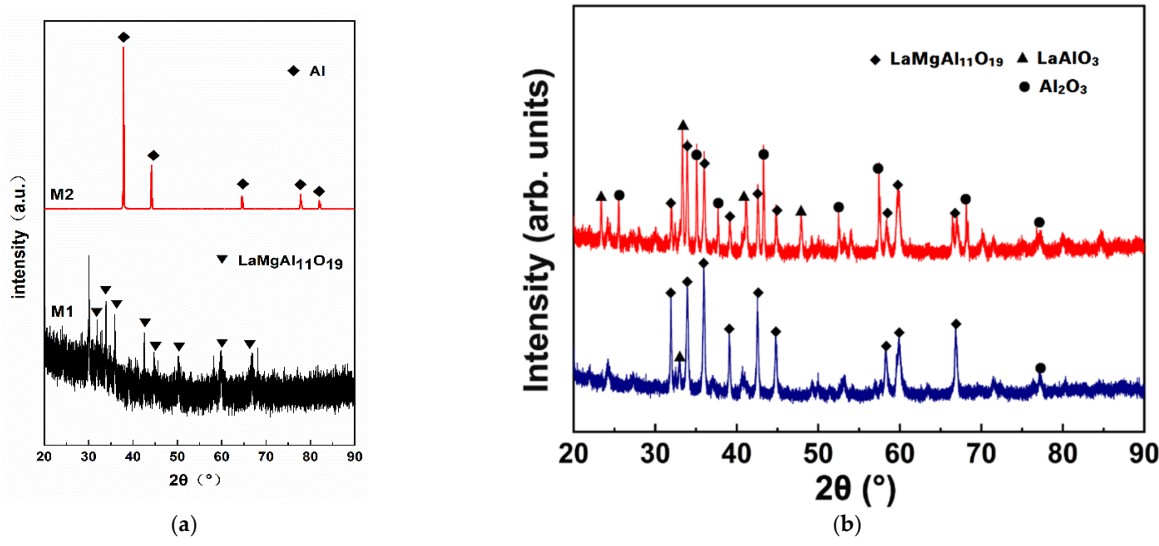

**Figure 13.** XRD results of Al layer (**a**) before and (**b**) after oxidation test at 1200 °C.

## 4. Conclusions

This study investigated the enhanced protection mechanism of arc aluminum coatings on CoCrNiAlY–YSZ–LMA double-layer ceramic coatings. Finite element analysis was performed to understand the distribution of the thermal stress of the coating. Combined with the experimental research results, the following conclusions can be drawn:

1.  In the absence of aluminum plating, the surface of the LMA layer of the CoCrNiAlY–YSZ–LMA double-layer ceramic coating had a large radial thermal stress (tensile stress, the direction is perpendicular to Y), and it increased with an increase in the operating temperature. This stress was caused by the volume shrinkage of the coating and was the main cause of the initiation and propagation of cracks in the axial direction (Y direction).

2.  The aluminum plating on the coating surface could effectively inhibit the volume shrinkage of the LMA layer through the good adhesion of the aluminum layer to the LMA, thereby considerably reducing the all-directional thermal stress on the surface of the LMA layer, preventing the initiation of axial microcracks, and protecting the coating. However, along the downward direction of the coating thickness, the protective effect of the aluminum coating gradually decreased, and the volume shrinkage of the LMA layer increased, which promoted radial (X-direction) microcracks inside the LMA layer.

3.  Aluminum plating on the surface of the coating can effectively bond the side walls of cracks by filling axial cracks, eliminating volume shrinkage, and eliminating all-directional thermal stress at the side walls of the cracks. Thus, it effectively inhibited further expansion of the axial cracks, showing good self-healing performance of the aluminum coating, and an enhancement effect on the coating. Moreover, the diffusion and adhesion of the aluminum-plated layer in the axial cracks effectively prevented the diffusion of oxygen to the inside of the coating through the cracks, reduced the rapid growth of the TGO layer, and inhibited the interface fracture. However, there was still a certain amount of shear thermal stress at the top and bottom of the crack (in the negative direction of the Y-axis), which become a dangerous point for the failure of the aluminum coating.

**Author Contributions:** Conceptualization, J.X.; Data curation, Z.W.; Formal analysis, J.X.; Funding acquisition, Y.F.; Investigation, S.H. (Shuai Hu); Methodology, J.X.; Project administration, Z.X.; Resources, Y.F., Y.C. and Z.X.; Software, J.X. and S.H. (Shuai Hu); Supervision, Y.F. and Y.C.; Validation, Z.W. and Z.X.; Visualization, S.H. (Suying Hu); Writing—original draft, J.X.; Writing—review & editing, Y.F. and Y.C. All authors have read and agreed to the published version of the manuscript.

**Funding:** This study was supported by the Southwest Institute of Technology and Engineering Cooperation fund (HDHDW5902020103), University of Science and Technology Liaoning Talent Project Grants (601011507–07).

**Institutional Review Board Statement:** Not applicable.

**Informed Consent Statement:** Not applicable.

**Data Availability Statement:** Data is contained within the article.

**Acknowledgments:** We sincerely thank the Beijing Power Machinery Institute for the coating spraying.

**Conflicts of Interest:** The authors declare no conflict of interest.

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
