# Peer review of "Understanding the Enhanced Protective Mechanism of CoCrNiAlY–YSZ–LaMgAl11O19 Double-Ceramic Coating with Aluminum Plating"

_coatings, doi:10.3390/coatings11111312_

Round 1
Reviewer 1 Report
The manuscript written by Juanfei Xu, Zhiguo Wang, Shuai Hu, YongjunFeng, Suying Hu, Yongjun Chen and Zhiwen Xie presents mainly a simulation model for understanding of thermal stress induced in CoCrNi-AlY-YSZ-LaMgAl11O19 coating.
The simulation model is presented in detail as well as the results. The SEM analysis presented at section 3.3 clearly demonstrated the benefit of aluminium plating of the coatings.
A very important aspect in analyzing the thermal resistence of coatings is the time.
The authors do not give information about the evolution of the plated coating properties with the time.
Before publication of the present manuscript in Coatings Journal, the authors need to address the following issues:
: how the aluminium plated microcracks evolve in time (days, month, years)
-how respond the aluminium plated layers to multiple thermal stress?
-how evolve the microcracks after multiple thermal stress applied at different temperatures and for different temporal intervals.
Author Response
Thank you very much for your valuable comments. We have carefully considered your comments and made active revisions. Attached is our reply. Hope you can adopt them, thank you.

Reviewer 2 Report
Dear Authors,
you should add some explanations concerning the oxidation of Al, see the notes in the attached pdf.

Author Response
Thank you very much for your valuable comments. We have carefully considered your comments and made active revisions. The following are the specific revisions. Hope you can adopt them, thank you.
Point 1: In this chapter you should explain also the behavior of aluminum at high temperatures - some sentences about the oxidation kinetics to form Al2O3 and how the whole TBC system will work after this.
Response 1: Regarding the behavior of Al2O3 at high temperature, I have modified it in the paper. Please refer to the attachment for specific information. Thank you.
Point 2: Not necessary to list it like this. (Table 2)
Response 2: Regarding the coating information listed in Table 2, we believe that the modeling situation can be more intuitively reflected, so it is not deleted.
Point 3: Here will it be Al2O3. If your answer is NOT, you must explain it. If YES, some data here are not correct.(Table 3)
Response 3: I’m sorry that we did not quote the references well for the Al2O3 parameters in Table 3. In fact, the author has a good integration of the previous Al2O3 parameters in the references, so I directly quoted his parameter, but the mistake was not cross-referenced in the previous submission of the manuscript. This is our negligence. Hope you can forgive me.
Point 4: always GPa. (Table 4 and others)
Response 4: For this question, I have re-planned the table in the manuscript and hope to adopt it, thank you.
Point 5: Why dont start with 1?
Response 5: Sorry for the typesetting error in the Conclusions, I have re-typeset in the manuscript and hope to adopt it, thank you.
Point 6: Improve and unify this: After how many authors is et.al?Are surnamer first? With CAPITAL letters or not? Chemical symbols.
Response 6: Sorry for the typesetting error in References. I have re-typeset in the manuscript and hope to adopt it. Thank you.
Because only one attachment can be uploaded, I am very sorry to answer your question in this way, and hope you can forgive me.

Reviewer 3 Report
The presented manuscript is devoted to the study of double ceramic coatings on the GH199 alloy. The manuscript is well organized and leaves a positive impression. To compare the experimental results and modeling, the authors used the finite element method. The article contains minor inaccuracies that need to be corrected before publication.
1. I suggest that the authors describe in more detail the coating process, including technological modes.
2. In Figure 1 there is an error in writing LaMgAl11O19.
3. I also recommend that the authors provide an XRD analysis of the upper surface of aluminum-coated samples after thermal tests. Does the formation of new compounds or evaporation of aluminum occur during thermal treatment? I think this point should be discussed in the manuscript.
Author Response
Thank you for your valuable comments. We have carefully considered your comments and made active revisions. The following are the specific revisions. Hope you can adopt them, thank you.
1. I suggest that the authors describe in more detail the coating process, including technological modes.
Point 1:I have modified this question in the manuscripts "2.1 Coating Preparation" and "2.3. Experimental Study", I hope it can be adopted, thank you.
2. In Figure 1 there is an error in writing LaMgAl11O19.
Point 2:I have modified this question in the manuscript "Figure 1", I hope it can be adopted thank you.
3. I also recommend that the authors provide an XRD analysis of the upper surface of aluminum-coated samples after thermal tests. Does the formation of new compounds or evaporation of aluminum occur during thermal treatment? I think this point should be discussed in the manuscript.
Point 3: I have modified this question in the manuscript "3.3. Experimental Analysis", I hope it can be adopted, thank you.
